# ConsintBench: Evaluating Language Models on Real-World Consumer Intent Understanding

## Abstract

Understanding human intent is a complex, high-level task for large language models (LLMs), requiring analytical reasoning, contextual interpretation, dynamic information aggregation, and decision-making under uncertainty. Real-world public discussions, such as consumer product discussions, are rarely linear or involve a single user. Instead, they are characterized by interwoven and often conflicting perspectives, divergent concerns, goals, emotional tendencies, as well as implicit assumptions and background knowledge about usage scenarios. To accurately understand such explicit public intent, an LLM must go beyond parsing individual sentences; it must integrate multi-source signals, reason over inconsistencies, and adapt to evolving discourse, similar to how experts in fields like politics, economics, or finance approach complex, uncertain environments. Despite the importance of this capability, no large-scale benchmark currently exists for evaluating LLMs on real-world human intent understanding, primarily due to the challenges of collecting real-world public discussion data and constructing a robust evaluation pipeline. To bridge this gap, we introduce CONSINT-BENCH, the first dynamic, live evaluation benchmark specifically designed for intent understanding, particularly in the consumer domain. CONSINT-BENCH is the largest and most diverse benchmark of its kind, supporting real-time updates while preventing data contamination through an automated curation pipeline. We evaluate 20 LLMs, spanning both open-source and closed-source models, across four core dimensions of consumer intent understanding: *depth*, *breadth*, *informativeness*, and *correctness*. Our benchmark provides a comprehensive and evolving evaluation standard for assessing LLM performance in understanding complex, real-world human intent, with the ultimate goal of advancing LLMs toward expert-level reasoning and analytical capabilities.

## 1 Introduction

The advent of Large Language Models (LLMs) OpenAI (2025); Grattafiori et al. (2024); Guo et al. (2025) has fundamentally transformed artificial intelligence, shifting from text generation to the ability to understand and reason about complex, real-world human intent Team (2025). These models demonstrate exceptional performance across a wide range of tasks Brown et al. (2020); Ouyang et al. (2022); Achiam et al. (2023); Chowdhery et al. (2023); Touvron et al. (2023); Google (2024). To assess LLMs in real-world problem-solving contexts, several benchmarks have been proposed. For example, SWE-bench Jimenez et al. (2024b) evaluates LLMs' ability to resolve software issues using GitHub repositories, while SPIDER2.0 Lei et al. (2024) focuses on enterprise-level text-to-SQL workflows. GAIA Mialon et al. (2023) introduces multi-modal, tool-augmented queries that require reasoning and web-browsing capabilities, and FutureX Zeng et al. (2025) challenges LLMs with future event prediction tasks. These benchmarks reflect a growing trend toward evaluating LLMs in dynamic, context-rich environments, aligning with more complex real-world applications.

Despite these advances, the question of whether LLMs truly understand public, swarm-like intent intelligence and the deeper, abstract aspects of human intent remains largely unexplored. Real-world human perspectives, whether in consumer decision-making, team collaboration, or online community discussions, are inherently multifaceted. They involve not only knowledge comprehension but

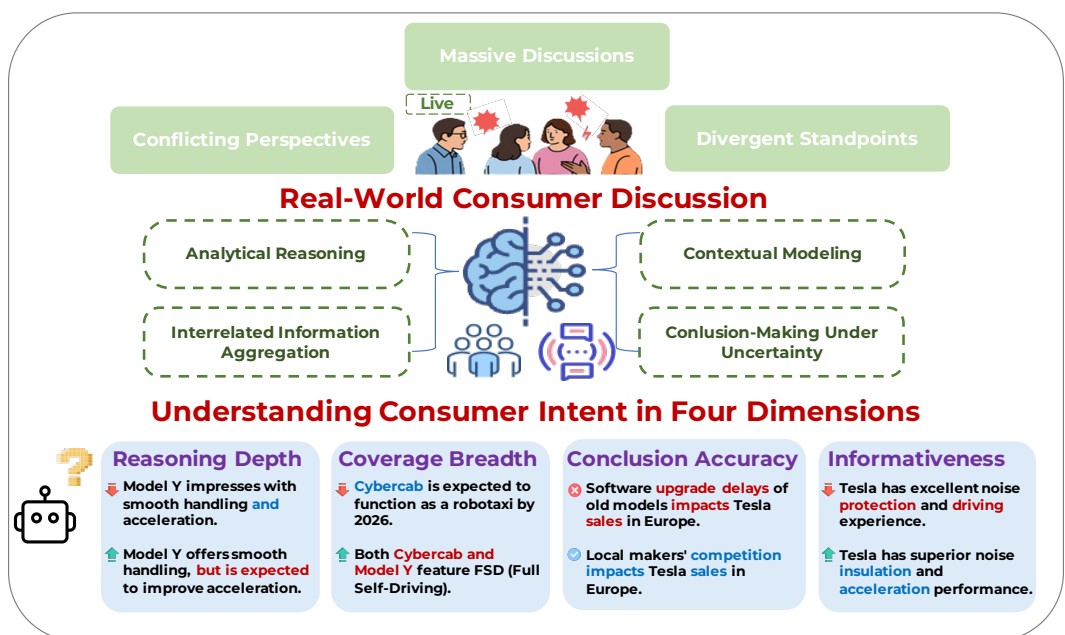

Figure 1: Overview of CONSINT-BENCH, a large-scale and live benchmark designed to evaluate real-world consumer intent understanding.

also a complex interplay of perspectives, needs, emotions, and implicit assumptions. Existing work often simulates individual user preferences but fails to account for the complex interactions of intentions, perspectives, and emotions across multiple users. To address this, a more comprehensive evaluation is needed—one that goes beyond individual perspectives to synthesize and aggregate intentions from multiple users. This requires the ability to map structured intention graphs that capture the multifaceted nature of human discourse.

Several benchmarks have focused on specific aspects of human intent, such as SocialIQA Sap et al. (2019) for social intent and commonsense reasoning, and TOMI Le et al. (2019) for Theory of Mind capabilities. However, these benchmarks rely on hand-crafted or semi-synthetic data, lacking the noise, redundancy, and subtext inherent in real-world discussions, which leads to evaluation processes that do not align with actual application scenarios. IFEVAL Zhou et al. (2023) evaluates instruction-following ability, while SociaBench Chen et al. (2024a) and AgentSense Mou et al. (2025) focus on intent understanding, generation quality, and social intent navigation. Emotion-Queen Chen et al. (2024b) addresses implicit emotions in human intent, and URS-bench Wang et al. (2024) evaluates LLMs' responses to factual question answering, problem-solving, and advice. However, these frameworks mainly focus on lower-level aspects of human intent, such as instruction-following, social reasoning, or emotional understanding, and lack an analysis of deeper dimensions of intent. In real-world scenarios, human intent is multifaceted and dynamic, involving social, emotional, and practical factors that require the fusion and resolution of conflicting viewpoints—key components of human reasoning. Existing frameworks, however, fail to evaluate these primary dimensions of LLM performance.

To address these limitations, we propose CONSINT-BENCH, a comprehensive, dynamic, and live benchmark designed to evaluate LLM performance in understanding real-world human intent, particularly in consumer domains. CONSINT-BENCH spans nine primary consumer domains, ranging from personal care and AI products to daily necessities, covering 54 sub-categories and over 1,400 product discussions sourced from real-world user interactions. For each product, we collect approximately 200 user comments, aggregating over 200k opinions. We evaluate LLMs' intent understanding ability across four primary dimensions: depth, breadth, correctness, and informativeness. Depth is further defined across five hierarchical levels (L1–L5), where the first three levels capture content directly from user discussions, while the last two require the model to reason based on internal knowledge and context, necessitating a deeper understanding of human intent. Addi-

tionally, we construct a robust evaluation pipeline to ensure accurate assessment and mitigate LLM bias and hallucination. We conduct extensive experiments on a variety of LLMs, including both closed- and open-source models, as well as reasoning and general models. Our results reveal that reasoning models outperform general models in depth, breadth, and correctness. However, a significant gap remains between closed-source and open-source models. Furthermore, even the most advanced models struggle with deep and broad intent understanding, highlighting substantial room for improvement.

In summary, our contributions are as follows:

- We introduce CONSINT-BENCH, a large-scale benchmark for real-world consumer intent, consisting of over 200k product-level discussions spanning 9 major domains, 54 sub-domains, and 1400+ products. Each topic includes an average of 200 discussion entries, ensuring information density and diversity.

- We define four primary aspects and implement a robust evaluation pipeline to mitigate bias and hallucination. Specifically, we construct CONSINT-TREE to assess LLMs' depth and breadth of intent understanding, use CONSINT-RAG for evaluating correctness, and measure informativeness through lexical diversity and semantic richness.

- We conduct extensive experiments on both closed-source and open-source models of varying sizes (1.5B to 72B parameters). The results show that even the most advanced models struggle with deep and broad intent understanding, highlighting significant potential for improvement.

## 2 BENCHMARK CONSTRUCTION

### 2.1 DATA CURATION

To retrieve and organize discussions from diverse online sources, we construct an automated system for collecting consumer discussions. The data collection pipeline integrates three stages:

**Search.** We employ a combination of vector search and API search to maximize the retrieval of relevant discussions. Vector search analyzes the semantic similarity between the user's input and available discussions, retrieving those with a similarity score above a defined threshold. API search utilizes LLMs to generate relevant keywords from the user's input, enhancing the search process.

**Retrieving.** After the search, the system retrieves relevant discussions from open-source web sources in real-time. These results are then used as raw input for the subsequent cleaning and filtering stages.

**Cleaning.** The collected results are filtered through both rule-based and LLM-based quality checks. First, discussions with titles and content shorter than 20 characters are discarded as low-quality. Second, discussions deemed irrelevant to the search topic by the LLM are excluded. Additionally, the cleaning process incorporates recency by considering time-based factors to ensure the relevance of the discussions.

This multi-step pipeline efficiently collects high-quality, contextually relevant discussions, ensuring the data is well-suited for subsequent analysis.

### 2.2 DATA STATISTICS

As shown in Table 1 and Figure 2, CONSINT-BENCH spans 54 sub-categories and includes over 1,400 product discussions sourced from real-world discussions. For each product, we collect approximately 200 user comments, aggregating over 200k opinions across categories such as personal care, AI products, and daily necessities. This rich dataset forms a robust foundation for evaluating LLMs' ability to understand human intent in diverse real-world contexts.

### 2.3 DIMENSION CATEGORIES

To thoroughly evaluate the capabilities of LLMs in understanding consumer intent, we categorize the evaluation into four primary dimensions: depth, breadth, informativeness, and correctness:

Table 1: Comparison with existing related benchmarks. "Real-world" indicates whether the data is sourced from real-world scenarios rather than synthetic or online existing resources. "Live Update" denotes whether the benchmark can be regularly updated.

| Benchmark | Domain | Tasks | Real World | Live Update |
|---|---|---|---|---|
| DABstep Egg et al. (2025) | Data Science | 450 | ✓ | ✗ |
| FutureX Zeng et al. (2025) | Future Prediction | 500/week | ✓ | ✓ |
| GAIA Mialon et al. (2023) | General QA | 466 | ✓ | ✗ |
| OSWorld Xie et al. (2024) | Computer Use | 369 | ✓ | ✗ |
| OPT-Bench Li et al. (2025) | Iterative Optimization | 30 | ✓ | ✗ |
| Spider2.0 Lei et al. (2024) | Text-to-SQL | 632 | ✓ | ✗ |
| SWE-Bench Jimenez et al. (2024a) | Code | 2,294 | ✓ | ✗ |
| SociaBench Chen et al. (2024a) | Social Intent | 6,000 | ✗ | ✗ |
| URS-bench Wang et al. (2024) | Intent Understanding | 1,846 | ✗ | ✗ |
| CONSINT-BENCH (ours) | Consumer Intent | 1,475 | ✓ | ✓ |

**Depth** measures the model's ability to analyze and provide insights into a given discussion, specifically evaluating how well it can explore complex ideas and offer comprehensive explanations. We define five levels of depth (L1–L5), where L1–L3 represent basic understanding—such as identifying usage scenarios, discussing product aspects, and capturing the user's feelings, all of which can be directly derived from the user discussion. Levels L4–L5 represent advanced comprehension, involving tasks like making comparisons with previous versions or similar products, analyzing their advantages and disadvantages, and speculating on the product's future direction. A higher depth score indicates a more profound and comprehensive understanding of the topic.

**Breadth** evaluates the model's ability to address a wide range of subtopics within a broader subject area. This dimension focuses on the model's versatility and capacity to cover various aspects of the topic, such as different usage scenarios, product versions, and product features. A higher breadth score indicates a more comprehensive understanding of the topic, as the model effectively spans a wider array of related issues.

**Informativeness** evaluates how effectively the model conveys content while maintaining its core message, reflecting its ability to provide meaningful information without unnecessary repetition or relying on a single paradigm. We assess informativeness through lexical richness and semantic redundancy, measuring the model's capacity to eliminate redundant or repetitive information. A lower redundancy score indicates a more focused and informative understanding.

**Correctness** evaluates whether the LLM's understanding of consumer intent is free from bias or hallucinations, and whether the responses accurately reflect the true opinions and sentiments expressed in the original discussions. A higher correctness score indicates a more accurate and reliable response, ensuring that the model's output aligns closely with the original human intentions.

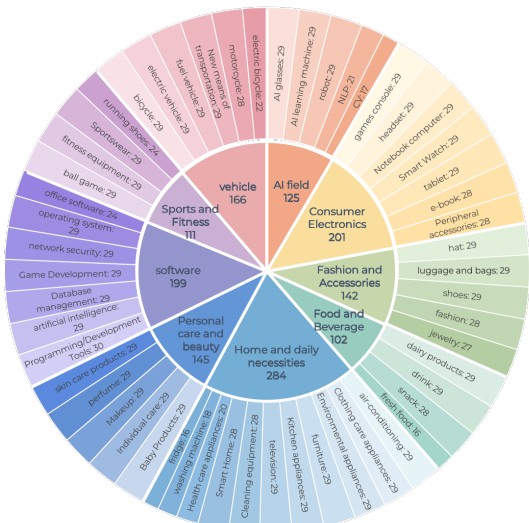

Figure 2: Overview of CONSINT-BENCH: It includes over 200k product-level discussions across 9 major domains, 54 sub-domains, and more than 1,400 products.

## 3 METHODOLOGY

To ensure a fair and actionable evaluation process, we adopt a paradigm where the LLM self-generates both the question and the answer. The question represents a summarized intent from

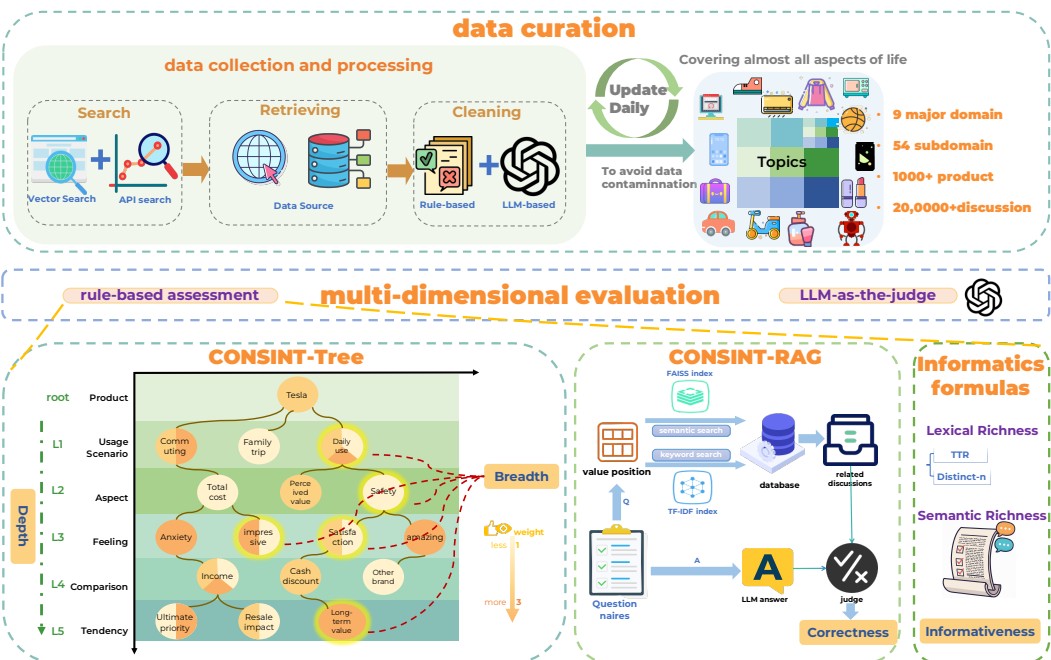

Figure 3: The overall pipeline of CONSINT-BENCH, covering data curation and evaluation. Data Curation: We provide two methods (keyword and semantic) to search and retrieve product discussions daily. It consists of over 200k product-level discussions across 9 major domains, 54 subdomains, and more than 1,400 products. The benchmark focuses on four primary dimensions and evaluates intent depth across five hierarchical difficulty levels. Afterthat, rule-based and LLM-based filtering is applied to remove irrelevant discussions and retain relevant ones. Evaluation: We propose CONSINT-TREE for more accurate assessment of depth and breadth dimensions, and CONSINT-RAG for correctness. For informativeness, we assess lexical richness and semantic redundancy.

the original discussion, and the answer reflects the LLM's judgment of the intent. This approach is repeated with the same number of questions to assess the model's ability to capture the most valuable majority perspective. To comprehensively assess the ability of LLMs to understand complex consumer intentions, we propose a robust evaluation methodology that combines rule-based analysis with the LLM-as-a-judge mechanism for multi-dimensional evaluation.

**CONSINT-TREE: Depth and Breadth Evaluation.** CONSINT-TREE is a tree-structured knowledge graph derived from real-world consumer discussions. Each question in the generated questionnaire is mapped to a corresponding node in CONSINT-TREE, forming a subtree. The size and structure of this subtree quantify the breadth and depth of the LLM's understanding.

**CONSINT-RAG: Correctness Evaluation.** CONSINT-RAG is a retrieval-augmented generation pipeline designed to mitigate hallucinations and bias caused by the noisy nature of real-world discussions. Each questionnaire question is paired with a reference answer, and the CONSINT-RAG pipeline verifies the accuracy of these answers, assessing the correctness of the LLM's intent comprehension.

**Informativeness Evaluation.** To assess informativeness, we compute the lexical richness and semantic redundancy of the generated questionnaire. These metrics capture the diversity and specificity of the LLM's expressions, reflecting the richness of its understanding of consumer intent.

### 3.1 CONSINT-TREE CONSTRUCTION AND EVALUATION

**Construction** To comprehensively evaluate the ability of LLMs to understand large-scale real-world data, which contains a massive volume of similar and conflicting content, in both depth and breadth, we propose the CONSINT-TREE —a five-level weighted hierarchical tree with weights based on discussion popularity. As illustrated in Figure 3, the root node represents the focal product under

discussion. In terms of depth, nodes across Levels 1–3 capture the product's usage scenarios, aspects, and user experience (e.g., usage feelings) as reflected in consumer discussions. Levels 4–5 deepen the understanding of consumer intent: Level 4 nodes represent competing products influencing consumer sentiments, while Level 5 nodes indicate potential improvement tendencies derived from these sentiments. In the CONSINT-TREE, a path from the root node to a leaf node defines a "branch," representing a progressively deepening user opinion. In terms of breadth, sufficient number of branches represent the multiple facets of the product's discussion. As for discussion popularity, nodes corresponding to high-frequency discussions or content with high upvotes/views are assigned higher weights. See the Appendix for detailed extraction of branches from each consumer discussion and the details of how related discussions are aggregated into high-weight nodes.

**Evaluation** To assess the depth and breadth of a LLM's understanding of consumer intent, content from questionnaires will be extracted into branches to lighten to CONSINT-TREE. Each branch will undergo semantic matching with the nodes in the CONSINT-TREE from top to bottom using a Sentence Transformer. Nodes that are successfully matched will be marked as "lightened," and the lightened nodes in the CONSINT-TREE will form a subtree. For the depth dimension, the depth score at each level (from L1 to L5) is calculated as the percentage of the total weight of lightened nodes in the subtree at that level relative to the total weight of all nodes in the original CONSINT-TREE at the same level. The overall depth score is computed as the average of the depth scores across all five levels. For the breadth dimension, the breadth score as the sum of the weights of all lightened nodes in the subtree. A higher depth score indicates that the questions in the questionnaire can delve into more profound layers of the comsumers intent. A higher breadth score reflects a more comprehensive understanding of the consumers intent.

## 3.2 CONSINT-RAG CONSTRUCTION AND EVALUATION

**Construction** To accurately evaluate the correctness dimension of LLMs in understanding consumer intent while mitigating LLM judge bias and hallucination, we propose CONSINT-RAG. This approach retrieves consumer preferences related to the LLM's inferred intent from the original discussions to serve as the ground truth. CONSINT-RAG follows a two-stage process: embedding and retrieval. In the embedding stage, user discussions are transformed into vector representations using TF-IDF and all-MiniLM-L6-v2. This dual representation enables the retrieval process to capture both precise keyword-level matches and deeper semantic information. In the retrieval stage, keywords are extracted from the LLM's questionnaire questions. These extracted key opinions and full questions are vectorized and jointly searched across two vector databases to retrieve the top-k most relevant discussions.

**Evaluation** After retrieval, the top-k most relevant discussions are analyzed to reflect human opinions and compared to the answers provided by the LLM during questionnaire generation. However, due to the implicit, noisy, and multi-opinion nature of real-world discussions, direct answer matching is not feasible. Therefore, further reasoning is required to determine the consensus opinion, reflecting the majority perspective. Based on this reasoning, the final answer is generated from the previous RAG results. The accuracy of the LLM's original answers for a given questionnaire is then used as the correctness evaluation metric.

## 3.3 INFORMATIVENESS

To evaluate the informativeness of LLMs in understanding consumer intent, we quantify Lexical Richness and Semantic Redundancy using informatics formulas.

**Lexical Richness:** Lexical richness is assessed across two dimensions: words and phrases. It reflects the LLM's ability to capture a broader range of consumer intent topics and diverse question formulations, thereby demonstrating a more comprehensive understanding of intent. Additionally, a more precise and nuanced expression of intent contributes to greater lexical richness. Type-Token Ratio (TTR) Johnson (1944) is used to evaluate word richness, while Distinct-n Li et al. (2016) measures phrase richness. Detailed metric calculations are provided in Appendix A.3.

**Semantic Redundancy:** Semantic redundancy is evaluated by assessing the embedded vector similarity between questionnaire questions, which helps gauge the LLM's ability to identify and structure consumer intent effectively. High semantic redundancy within the questionnaire indicates that the

Table 2: Performance of reasoning LLMs, general LLMs, and open-source LLMs on CONSINT-BENCH, with the best performance highlighted in **bold**.

| Model | Depth | | | | | | Breadth | Informativeness | | Correctness |
|---|---|---|---|---|---|---|---|---|---|---|
| | L1 | L2 | L3 | L4 | L5 | Overall | | Lexical | Semantic↓ | |
| *Proprietary LLMs* | | | | | | | | | | |
| GPT-5 | 18.29 | **25.81** | **6.25** | **3.49** | 0.06 | 10.78 | **53.48** | 80.21 | 62.75 | 62.65 |
| GPT-4.1 | 20.97 | 25.03 | 5.90 | 3.37 | 0 | 11.01 | 53.41 | 79.07 | 63.82 | 59.05 |
| GPT-4o | 20.99 | 23.70 | 4.90 | 2.58 | 0.05 | 10.44 | 52.95 | 79.56 | 62.86 | 75.75 |
| Claude-3.5-sonnet | 19.25 | 23.25 | 5.34 | 2.95 | 0 | 10.16 | 52.83 | 73.94 | 61.11 | 53.35 |
| GPT-o3 | 16.17 | 22.43 | 5.69 | 3.18 | **0.07** | 9.51 | 52.73 | **85.52** | **52.27** | **80.35** |
| *Open-Source LLMs* | | | | | | | | | | |
| Qwen3-30B-A3B | **25.01** | 23.66 | 5.43 | 2.51 | 0.06 | **11.33** | 53.20 | 70.58 | 68.94 | 61.60 |
| DS-Distill-Qwen-14B | 17.00 | 25.56 | 6.12 | 3.30 | 0 | 10.40 | 53.32 | 67.47 | 75.53 | 58.45 |
| Qwen2.5-32B-Instrcut | 19.26 | 23.63 | 5.31 | 2.89 | 0.01 | 10.21 | 52.46 | 65.72 | 74.60 | 54.95 |
| Qwen3-32B | 20.77 | 23.46 | 5.57 | 2.82 | 0 | 10.52 | 51.15 | 65.84 | 68.16 | 55.26 |
| Qwen3-8B | 15.95 | 22.43 | 4.89 | 2.39 | 0.01 | 9.13 | 50.58 | 57.51 | 81.25 | 50.42 |
| Qwen2.5-72B-Instrcut | 18.89 | 22.27 | 5.73 | 3.10 | 0 | 10.00 | 50.52 | 54.63 | 77.49 | 64.11 |
| DS-Distill-Qwen-32B | 16.60 | 24.56 | 5.90 | 3.30 | 0 | 10.07 | 50.34 | 59.30 | 76.58 | 53.90 |
| Qwen2.5-14B-Instrcut | 13.56 | 22.23 | 5.45 | 2.87 | 0.02 | 8.83 | 48.27 | 52.39 | 80.06 | 60.88 |
| LLama3.2-8B-Instrcut | 13.88 | 19.75 | 5.62 | 2.73 | 0 | 8.40 | 47.91 | 47.87 | 88.25 | 52.31 |
| Qwen2.5-7B-Instrcut | 11.87 | 19.73 | 4.16 | 1.97 | 0 | 7.54 | 47.43 | 43.58 | 85.07 | 49.24 |
| Internlm3-8B-Instrcut | 11.07 | 20.76 | 4.87 | 2.61 | 0.03 | 7.87 | 45.91 | 49.83 | 75.51 | 51.67 |
| LLama3.1-8B-Instrcut | 11.23 | 19.46 | 5.53 | 2.91 | 0 | 7.83 | 45.41 | 42.36 | 88.00 | 52.67 |
| Qwen2.5-3B-Instrcut | 13.49 | 18.63 | 4.22 | 2.09 | 0 | 7.69 | 42.73 | 39.32 | 79.35 | 35.43 |
| Qwen2.5-1.5B-Instrcut | 2.83 | 4.94 | 0.99 | 0.45 | 0 | 1.84 | 14.31 | 4.56 | 87.65 | 36.90 |
| DS-Distill-Qwen-7B | 1.80 | 4.91 | 1.35 | 0.55 | 0 | 1.72 | 11.54 | 3.30 | 73.25 | 13.40 |

LLM's logical reasoning approach may be overly simplistic or that it has failed to capture the full spectrum of consumer intent. Redundancy is calculated as the average maximum similarity between each question and all other questions Chen et al. (2021b). Detailed metric calculations are provided in Appendix A.3.

# 4 EXPERIMENT

## 4.1 EXPERIMENTAL SETUP

We evaluated our method across a diverse set of LLMs, including both proprietary and open-source models, each consisting of reasoning and general models. The proprietary models include OpenAI's GPT family and Claude, all accessed via their APIs. For open-source models, we consider the Qwen series (ranging from 1.5B to 72B), LLaMA, DeepSeek and InternLM, all deployed locally using the LMDeploy framework.

## 4.2 MAIN RESULTS

Table 2 evaluates the performance of 20 LLMs in understanding consumer intent across four key dimensions:

**1) Depth:** The scores across L1–L5 generally show a downward trend, reflecting the increasing difficulty of capturing deeper aspects of consumer intent. GPT-5 and GPT-4.1 achieve the highest overall depth scores, ranking first and second, respectively, in L2 and L3, demonstrating a comprehensive understanding of both contextual elements and user sentiment. As a reasoning model, GPT-o3 excels in L5, highlighting the role of reasoning in deepening the understanding of consumer intent. Among open-source LLMs, Qwen3-30B-A3B, a Mixture of Experts (MOE) model, performs best, benefiting from its ability to allocate specialized experts for different depths of understanding.

**2) Breadth:** GPT-5 leads in breadth, showing its ability to address a wide range of consumer intent. In open-source models, Deepseek-R1-Distill-Qwen-14B demonstrates strong coverage of diverse subtopics. However, smaller models such as Qwen2.5-1.5B-Instruct struggle to capture the full breadth of consumer intent.

Table 3: Comparison of reasoning LLMs, general LLMs, and open-source LLMs using CONSINT-TREE on CONSINT-BENCH, with the best performance highlighted in **bold**.

| Model | Depth | | | | | | Breadth | Informativeness | | Correctness |
|---|---|---|---|---|---|---|---|---|---|---|
| | L1 | L2 | L3 | L4 | L5 | Overall | | Lexical | Semantic↓ | |
| GPT-o3 (wo/Tree) | 16.17 | 22.43 | 5.69 | 3.18 | 0.07 | 9.51 | 52.73 | 85.52 | **52.27** | **80.35** |
| GPT-o3 (w/Tree) | **45.13** | **36** | **15** | **11.74** | **1.37** | **21.95** | **59.16** | **72.47** | 71.05 | 57.10 |
| | (+28.96) | (+13.57) | (+9.31) | (+8.56) | (+1.30) | (+12.44) | (+6.43) | (-13.05) | (-0.22) | (-23.25) |
| GPT-4o (wo/Tree) | 20.99 | 23.70 | 4.90 | 2.58 | 0.05 | 10.44 | 52.95 | 79.56 | 62.86 | 75.60 |
| GPT-4o (w/Tree) | 37.38 | 31.39 | 12.78 | 9.60 | 0.85 | 18.40 | 57.00 | 70.77 | 72.36 | 64.15 |
| | (+16.39) | (+7.69) | (+7.88) | (+7.02) | (+0.80) | (+7.96) | (+4.05) | (-8.79) | (-0.50) | (-11.45) |
| Qwen2.5-7B (wo/Tree) | 11.87 | 19.73 | 4.16 | 1.97 | 0 | 7.54 | 47.43 | 43.58 | 85.07 | 42.15 |
| Qwen2.5-7B (w/Tree) | 32.39 | 29.78 | 11.29 | 7.96 | 0.54 | 16.39 | 49.28 | 42.33 | 79.63 | 49.24 |
| | (+20.52) | (+10.05) | (+7.13) | (+6.00) | (+0.54) | (+8.85) | (+1.85) | (-1.25) | (-5.44) | (+7.09) |

**3) Informativeness:** GPT-o3 outperforms all models in lexical richness and minimal semantic redundancy, indicating that its deeper understanding of consumer intent is supported by a broader vocabulary and refined semantic expression. Open-source reasoning models, while competitive in depth and breadth, generally lag behind proprietary models in lexical and semantic richness. Additionally, compared to reasoning open-source models, general open-source models such as Qwen2.5-72B-Instruct still struggle with understanding intent.

**4) Correctness:** GPT-o3 achieves the highest correctness score, underscoring the superior ability of reasoning-focused models to accurately summarize and derive consumer intent from large-scale discussions.

In summary, smaller open-source general models tend to underperform across all four dimensions, particularly in deeper reasoning (L5 depth) and expressive capabilities. Reasoning LLMs, with their advanced reasoning abilities, outperform in multiple metrics, emphasizing the critical role of reasoning in improving the depth, breadth, and correctness of consumer intent comprehension.

## 4.3 ABLATION STUDY

We conducted additional experiments to explore whether noisy and low-quality information in large-scale real-world discussions limits the understanding capabilities of large language models (LLMs). In these experiments, we replace the original real-world discussions with CONSINT-TREE for LLM evaluation. The results are presented in Table 3.

In the *Depth* and *Breadth* dimensions, the `LLM (w/Tree)` outperformed the `LLM (wo/Tree)`. This indicates that the `tree` significantly enhances the LLM's ability to understand high-weight, high-interest consumer intents, improving the overall depth and breadth of intent comprehension. However, in the *Informativeness* dimension, a decline was observed when comparing `LLM (w/Tree)` to `LLM (wo/Tree)`. This suggests that when LLMs process the refined intents extracted from the `tree`, their ability to understand these intents in a nuanced manner is constrained. This may be due to the fact that high-weight branches often focus on more popular aspects, leading to a reduction in semantic richness and overall informativeness. In the *Correctness* dimension, `LLM (w/Tree)` demonstrated a decreasing trend. Although the `tree` refines noisy and irrelevant discussions, it may simultaneously lose some critical information, resulting in a failure to provide a comprehensive representation of the relevant topics. In contrast, for open-source small models, `LLM (w/Tree)` effectively reduces noise in the consumer intents, leading to improved understanding correctness. This suggests that small models are more sensitive to real-world noise and may struggle to properly understand human intent in noisy environments.

## 4.4 FURTHER DISCUSSION

The results in Table 2 show that open-source reasoning models such as Qwen3-30B-A3B outperform close-source model in both depth and breadth. To explore this further, we conduct a case study that presents the performance of GPT-5, GPT-o3, and Qwen3-30B-A3B on CONSINT-BENCH in understanding consumer intent, specifically from discussions about the Google Nest Smart Speaker. GPT-5 achieves the highest breadth score by covering more high-weight nodes, while GPT-o3 uniquely excels in L5 depth. Although the breadth scores for all three models are comparable, Qwen3-30B-

A3B lags notably in informativeness. Notably, the question stems and options in Qwen3-30B-A3B's questionnaires are longer on average compared to those generated by GPT-5 and GPT-o3. This suggests that closed-source LLMs tend to use more refined and precise vocabulary and semantic structures when understanding consumer intent, highlighting their superior control over finer details. The detailed results are shown in A.4.

## 5 RELATED WORK

**LLM Evaluation** The rapid advancements in Large Language Models (LLMs) have led to the creation of numerous benchmarks to evaluate their generalization and reasoning capabilities. Early efforts, such as MMLU Hendrycks et al. (2020) and BIG-bench Srivastava et al. (2022), provided broad assessments of general knowledge and reasoning skills. Subsequent benchmarks focused on more specific domains, including linguistic and commonsense reasoning (e.g., GLUE Wang et al. (2018), SuperGLUE Wang et al. (2019), CommonsenseQA Talmor et al. (2019), HellaSwag Zellers et al. (2019), TruthfulQA Lin et al. (2022)), mathematical and programming reasoning (e.g., MATH Hendrycks et al. (2021), GSM8K Cobbe et al. (2021), HumanEval Chen et al. (2021a), MBPP Austin et al. (2021)), and task-based agent evaluation (e.g., MLE-bench Chan et al. (2024) for ML engineering, OSWorld Xie et al. (2024) for GUI tasks, and OPT-BENCH Li et al. (2025) for complex optimization). Despite these advancements, there remains a lack of systematic evaluation regarding whether LLMs can effectively understand human intent in dynamic, real-world decision-making contexts—particularly those involving multi-user perspectives, emotional nuance, and evolving goals. To address this gap, we introduce CONSINT-BENCH, a large-scale benchmark designed to evaluate LLMs' ability to comprehend and reason about human intentions in complex, real-world scenarios.

**LLM Human Intent Evaluation** Human intent evaluation has increasingly focused on understanding human-centric intent in complex, dynamic real-world scenarios. Benchmarks such as SocialIQA Sap et al. (2019) have emphasized social intent and commonsense reasoning, while TOMI Le et al. (2019) evaluates LLMs' Theory of Mind capabilities. Several benchmarks have assessed LLMs' ability to understand and follow human intent. For example, IFEVAL Zhou et al. (2023) primarily evaluates instruction-following ability, while SociaBench Chen et al. (2024a) and AgentSense Mou et al. (2025) assess intent understanding, generation quality, and social intent navigation. EmotionQueen Chen et al. (2024b) focuses on evaluating implicit emotions in human intent, and URS-bench Wang et al. (2024) evaluates LLMs' responses to factual question answering, problem-solving, and advice. However, these frameworks typically focus on specific aspects of human intent, such as instruction-following, social reasoning, or emotional understanding. In real-world scenarios, human intent is often multifaceted and dynamic, involving a combination of social, emotional, and practical factors. As a result, no existing framework provides a comprehensive evaluation of whether LLMs can truly understand human reasoning and mental states. To fill this gap, we propose CONSINT-BENCH, a benchmark designed to evaluate LLMs' ability to understand dynamic and complex real-world human intent.

## 6 CONCLUSION

In this work, we propose CONSINT-BENCH, a comprehensive benchmark consisting of over 200k product-level discussions across 9 major domains, 54 sub-domains, and over 1,400 products, designed to evaluate the performance of Large Language Models (LLMs) in understanding real-world human intent, particularly within consumer domains. Our evaluation framework measures LLMs' ability to comprehend intent across four key dimensions: depth, breadth, correctness, and informativeness. We implement a robust evaluation pipeline to mitigate bias and hallucinations. Specifically, we construct CONSINT-TREE to assess LLMs' depth and breadth of intent understanding, use CONSINT-RAG for evaluating correctness, and measure informativeness through lexical diversity and semantic richness. Through extensive experiments on both closed-source and open-source models, we demonstrate that reasoning models outperform general models on average. However, significant gaps remain between closed-source and open-source models, and even the most advanced models struggle with deep and broad intent understanding. Our mission is to advance LLMs toward expert-level reasoning and improve their ability to understand complex real-world intent.

REPRODUCIBILITY STATEMENT

We adhere to the reproducibility guidelines outlined in the ICLR 2026 author guidelines. All data and code necessary to reproduce our results will be open-sourced and made available as soon as possible.

ETHICS STATEMENT

The CONSINT-BENCH dataset was constructed from public available websites, and all privacy-sensitive personal information has been removed during the data curation process. To mitigate the potential for technology misuse, the benchmark will be released under a restrictive license for academic research purposes only, explicitly prohibiting malicious applications.

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

# A  APPENDIX

## A.1  USE OF LARGE LANGUAGE MODELS

Large Language Models are used for grammar check and polishing in this paper.

## A.2  CONSINT-TREE CONSTRUCTION DETAILS

First, the LLM (GPT-4o) is utilized to extract branches from each consumer discussion. During this process, the model is prompted to summarize a user's discussion following the template: ⟨Product_series⟩, in the ⟨Usage Scenario⟩, its ⟨Aspect⟩, compared with ⟨Comparison⟩, gives consumers the ⟨Feeling⟩ perception, and the discussion suggests that ⟨Tendency⟩.

Next, the branches are used to construct the tree. All branches are connected to the tree root node, forming an initial tree. Sentence Transformers are then employed to merge semantically similar nodes layer by layer from the top down within the initial tree. During this process, nodes in the same layer that share the same parent node and are semantically similar are merged into one. The child nodes of each pre-merged node are then designated as the child nodes of the merged node. Additionally, the weight of the merged node is calculated as the sum of the weights of its child nodes. Ultimately, the fully merged tree is referred to as CONSINT-TREE, which will resemble the structure shown in Figure X. Nodes with higher weights will appear in the shallow layers of the tree; this is because the core topics of discussion (e.g., usage scenarios, aspects, feelings) often overlap across discussions from different consumers, and such high-weight nodes represent the aspects of the product that users focus on most.

This process enables the clear presentation of user discussion content in a tree structure while highlighting discussion hotspots. Meanwhile, by updating the discussion data and reconstructing CONSINT-TREE, we can analyze changes in child nodes under the same parent node between the two trees—thereby identifying users' immediate concerns and long-term strategic considerations.

After summarization, the key terms in the template are fetched to form a single branch. For the branches derived from one discussion, the initial weight of each node is equal, ranging from 1 to 3, and determined by the discussion's upvotes and view count. Notably, not every discussion can be summarized to fill all six key terms—more successfully filled key terms correspond to a longer branch path, which in turn reflects a more in-depth consumer intent. Next, all branches are connected to the tree root node, forming an initial tree. Sentence Transformers are then employed to merge semantically similar nodes layer by layer from the top down. During this process, nodes in the same layer that share the same parent node and are semantically similar are merged into one. The child nodes of each pre-merged node are then designated as the child nodes of the merged node. Additionally, the weight of the merged node is calculated as the sum of the weights of its child nodes. Ultimately, the fully merged tree is referred to as CONSINT-TREE. Nodes with higher weights will appear in the shallow layers of the tree; this is because the core topics of discussion (e.g., usage scenarios, aspects, feelings) often overlap across discussions from different consumers.

**Lighten the Tree:** To assess the depth and breadth of a LLM's understanding of consumer intent, content from questionnaires will be extracted into branches. Each branch will undergo semantic matching with the nodes in the CONSINT-TREE from top to bottom using a Sentence Transformer. Nodes that are successfully matched will be marked as "lightened," and the lightened nodes in the CONSINT-TREE will form a subtree. and the questionnaire will receive the score corresponding to that node.

Specifically, for each question in the questionnaire, the question stem and its four options will first be concatenated into four opinion statements. The LLM will then extract four branches from these four statements. These four branches will be matched with the nodes in the CONSINT-TREE from top to bottom—each branch will lighten a path and obtain a score based on the weights of the nodes along that path. The branch with the highest score for a given question will be used to "lighten" the CONSINT-TREE. Notably, nodes in the CONSINT-TREE cannot be repeatedly lightened by different questions. After iterating through all questions, the lightened nodes in the CONSINT-TREE will form a subtree.

## A.3 INFORMATIVENESS

**Lexical Richness:** The evaluation of lexical richness relies on two key metrics: Type-Token Ratio (TTR) Johnson (1944) and Distinct-n Li et al. (2016). TTR quantifies the ratio of unique tokens to the total number of words in the text. It is defined as:

$$\text{TTR} = \frac{\text{Count(unique token)}}{\text{Count(tokens)}}$$

where a higher TTR indicates greater lexical richness. Distinct-n focuses on the *n-gram* level, measuring the ratio of unique *n-grams* to the total number of *n-grams*. This study focuses on *bi-grams*, and the Distinct-n is calculated as:

$$\text{Distinct-n} = \frac{\text{Count(unique bi-gram)}}{\text{Count(bi-grams)}}.$$

**Semantic Redundancy** is evaluated using a self-referential manner Chen et al. (2021b), where the average maximum semantic similarity is computed between each question and all other questions in the questionnaire, as well as between each question's options and all other questions' options. Given a set of questions $Q = \{q_1, q_2, ..., q_n\}$, the semantic similarity between any two questions $q_i$ and $q_j$ is calculated using cosine similarity:

$$\text{Sim}(q_i, q_j) = \frac{\mathbf{v_i} \cdot \mathbf{v_j}}{\|\mathbf{v_i}\|\|\mathbf{v_j}\|},$$

where $\mathbf{v_i}$ and $\mathbf{v_j}$ represent the vector embeddings of questions $q_i$ and $q_j$, respectively. The redundancy score is then computed as the average of the maximum similarity values across all pairs of questions:

$$\text{Redundancy} = \frac{1}{n} \sum_{i=1}^{n} \max_{j \neq i} \text{Sim}(q_i, q_j).$$

Notably, a lower redundancy score indicates less repetition in question paradigms and option designs, which reflects the LLM's ability to understand consumers' intentions from multiple perspectives and conduct multi-source causal inference.

## A.4 CASE STUDY

Table 4: Performance of reasoning LLMs, general LLMs, and open-source LLM on Google Nest Smart Discussion.

| Model | Depth | | | | | | Breadth | Informativeness | | Correctness |
|---|---|---|---|---|---|---|---|---|---|---|
| | L1 | L2 | L3 | L4 | L5 | Overall | | Lexical | Semantic↓ | |
| GPT-5 | 3.33 | **21.79** | **6.53** | **4.19** | 0.00 | 7.17 | **50.84** | **0.87** | **0.30** | 0.75 |
| GPT-o3 | 2.51 | 20.02 | 5.33 | 2.77 | **1.91** | 6.51 | 50.19 | 0.75 | 0.46 | **0.95** |
| Qwen3-30B-A3B | **31.32** | 16.61 | 2.77 | 1.49 | 0.00 | **10.44** | 50.51 | 0.81 | 0.39 | 0.75 |

### A.4.1 GOOGLE NEST SMART SPEAKER QUESTIONNAIRE FROM GPT-5

1. **How do you primarily use your Google Nest speakers at home?**

    A. For music playback

    B. For controlling smart devices

    C. For asking questions/time/weather

    D. For security alerts or doorbell announcements

    *Answer: A. Users reported using Nest speakers most often for music, followed by smart home control and daily reminders such as weather or timers.*

2. **How satisfied are you with the sound quality of Nest Audio compared to Nest Mini?**

    A. Nest Audio is leagues better, especially bass

    B. Mini is enough for casual listening

    C. Nest Audio is adequate but not impressive

    D. No difference noticed

*Answer: A. Users consistently said Nest Audio has much better bass and overall quality compared to Mini, making it preferable for music.*

3. **Have you experienced connection issues with your Nest speakers in recent years?**

    A. Yes, frequent disconnections and 'sorry something went wrong'

    B. Yes, occasional hiccups

    C. No, they work reliably

    D. Issues only due to Wi-Fi provider/router

*Answer: A. Many users reported worsening connection reliability over time, though some fixed issues by upgrading Wi-Fi or resetting devices.*

4. **How well does Google Home integrate with your non-Google devices (like Tuya, Zig-Bee, or Ikea smart products)?**

    A. Very smooth integration

    B. Works but often buggy

    C. I cannot integrate them at all

    D. I only use 100% Google products

*Answer: B. Several users noted persistent issues integrating Tuya/Lidl ZigBee and Ikea products with Google Home compared to their native apps.*

5. **Which competing ecosystem do you feel has smoother smart home integration?**

    A. Apple HomeKit/HomePod

    B. Google Nest/Home

    C. Amazon Alexa/Echo

    D. None of them are ideal

*Answer: A. Some users with Apple devices felt HomePod integration was smoother, while Google or Amazon were preferred for value and compatibility.*

6. **What's your biggest frustration with Nest smart speakers?**

    A. Laggy responses

    B. Misunderstood commands

    C. Device going offline

    D. Limited software updates

*Answer: A. Users most commonly complained about slow responses and frequent 'didn't understand' errors, especially in recent years.*

7. **How often do you use your Nest devices for multi-room audio?**

    A. Daily

    B. Occasionally for parties/events

    C. Rarely

    D. Never

*Answer: A. Users said multi-room audio used to work seamlessly but has become frustratingly unreliable, with groups often not functioning.*

8. **Would you consider switching away from Google due to reliability issues?**

    A. Yes, to Alexa

    B. Yes, to Apple

    C. Yes, to Home Assistant or local-first systems

    D. No, I will stay with Google

*Answer: A. Some users said they were ready to move to Alexa or Home Assistant because of lag and reliability problems with Nest.*

9. **Do you find Nest speakers useful for families with children?**

    A. Yes, great for music and educational features
    B. Somewhat, but limited reliability frustrates kids
    C. Not useful at all for kids
    D. Only good as background music

    *Answer: B. Users mentioned features like 'animal of the day' were fun for kids, but white noise and music playback have become unreliable.*

10. **How important is price compared to reliability when choosing between Google, Amazon, or Apple smart speakers?**

    A. Price is the most important
    B. Balance between price and reliability
    C. Reliability is the most important
    D. Ecosystem integration matters most

    *Answer: A. Users often noted that Google's affordability got them locked in, but reliability issues make them consider pricier alternatives.*

11. **Have you faced limitations when casting media from phones to Nest devices?**

    A. Yes, apps don't always connect
    B. Sometimes there's lag/delay
    C. No issues at all
    D. I don't use casting

    *Answer: A. Users reported frequent casting issues, especially with Spotify, YouTube, and video streams not reaching certain Nest devices.*

12. **If you primarily wanted a device for music, which would you pick?**

    A. Google Nest Audio
    B. Amazon Echo 4th gen
    C. Apple HomePod Mini
    D. Traditional Bluetooth speakers

    *Answer: A. Users debated between Nest Audio, Echo, and HomePod. Many said Nest Audio had good bass but Echo was decent, while some still preferred Sonos or passive Bluetooth sets.*

13. **How do you feel about Google discontinuing/reducing stock of Nest devices?**

    A. Concerned about product support
    B. Neutral, waiting for new models
    C. Considering switching to another brand
    D. Not worried at all

    *Answer: A. Several users worried Nest Mini and Audio are discontinued, wondering if Google will abandon the smart speaker hardware.*

14. **What feature would make you more likely to stick with Google Nest speakers?**

    A. Improved reliability and faster response
    B. Better music/audio quality
    C. Deeper integration with third-party devices
    D. Clear roadmap and updates from Google

    *Answer: A. Users said better reliability, sound improvements, and smoother ecosystem updates would convince them to remain loyal.*

15. **Do you experience more issues with Google Assistant understanding you in multilingual households?**

    A. Yes, it constantly misinterprets
    B. Sometimes, especially switching languages
    C. No, it works fine in multiple languages

D. I only use one language

*Answer: A. Users noted Assistant struggles badly in multilingual homes, often failing basic commands or mixing languages incorrectly.*

16. **What's your perspective on Nest speakers' long-term durability?**

    A. Still working fine years later
    B. Performance has worsened over time
    C. Hardware is durable but software declines
    D. They feel like e-waste now

    *Answer: C. Some users praised durability, while many complained hardware outlasted software support, calling devices obsolete early.*

17. **How do you primarily resolve issues with Nest devices?**

    A. Factory reset
    B. Router and Wi-Fi upgrades
    C. Reinstalling Google Home app
    D. Contacting Google support

    *Answer: A. Most users resorted to factory resets or Wi-Fi upgrades; official support was rarely mentioned as helpful.*

18. **Would you invest in another Nest smart display (like Hub/Hub Max) now?**

    A. Yes, I still trust Google ecosystem
    B. Maybe, if I find a second-hand deal
    C. No, too much risk of discontinued support
    D. I prefer other brands' smart displays

    *Answer: C. Users were hesitant to buy discontinued Nest Hubs/Max, fearing bricking or lack of updates.*

19. **When connecting Nest with services like Spotify or YouTube Music, what's your experience?**

    A. Smooth, works well
    B. Works but occasionally lags
    C. Often breaks or blocks premium-only features
    D. I don't link music services

    *Answer: C. Several users reported Spotify on Nest sometimes says 'premium only' even with premium, and YouTube Music integration often fails.*

20. **What future direction should Google take with Nest smart speakers?**

    A. Bring Gemini AI with better natural understanding
    B. Focus on keeping devices reliable
    C. Produce new affordable hardware
    D. Open-source support if retiring devices

    *Answer: B. Users speculated Google must fix reliability, offer Gemini AI improvements, and either release new hardware or open source old devices.*

### A.4.2 GOOGLE NEST SMART SPEAKER QUESTIONNAIRE FROM GPT-O3

1. **When users retrofit 1980s intercoms with Nest Mini units, which room-specific control do they hope to achieve later on?**

   A. Only ceiling fans and lights of that bedroom
   B. Satellite TV channels in the garage
   C. Printer queues in the study
   D. Irrigation valves in the backyard

*Answer: A. Users describe planning "a speaker in every bedroom with some intricate setup to both only control devices specific to that room (like ceiling fans and lights) as well as shared devices."*

2. **What adjective did a long-time owner use to praise Nest Audio's stereo sound after pairing two units?**

    A. Incredible

    B. Tinny

    C. Muffled

    D. Overpriced

*Answer: A. A commenter said "I use 2 Nest Audios in a stereo setup, and the audio is incredible," reflecting positive feelings about sound quality.*

3. **Which competing smart speaker line did several Redditors say they might switch to because Google devices have become "laggy" and "driving me insane"?**

    A. Amazon Echo / Alexa

    B. Sonos Era

    C. Bose Smart Ultra

    D. Marshall Uxbridge

*Answer: A. Many posts mention considering Amazon Echo or Alexa displays as an alternative when Nest performance deteriorated.*

4. **In the thread about buying a Nest Hub Max, which security-related use case was highlighted as a reason to still want the display?**

    A. Acting as a digital photo frame with camera recording

    B. Hosting a VPN server

    C. Controlling sprinklers via Zigbee

    D. Calibrating 3D printers

*Answer: A. A buyer said they liked "the camera/security recording function and using it as a digital photo frame," showing the usage scenario.*

5. **Which phrase did frustrated owners repeatedly hear instead of successful commands, prompting them to call Google Home a "support group"?**

    A. "Sorry, something went wrong, try again later."

    B. "Firmware upgrade in progress."

    C. "Device is paired in another room."

    D. "Low battery, shutting down."

*Answer: A. Multiple users quote the device replying "Sorry, something went wrong, try again later," illustrating a common pain point.*

6. **Why did one user say the Pixel Tablet on its dock feels like an "old TV/VCR combo" compared with a real Nest Hub?**

    A. It can't be asked to play music on other Google speakers

    B. It lacks Wi-Fi 6E support

    C. The screen is smaller than 5 inches

    D. It forces Amazon Prime ads

*Answer: A. They complained that you "can't tell it to play music on it from another Google speaker," so the hybrid device does neither role well.*

7. **Which connectivity problem did a border-area listener report when TuneIn stations kept dropping on Nest speakers?**

    A. Occasional to frequent loss in signal

    B. Crackling Bluetooth interference only at night

    C. Wrong language playback

    D. Overheating power adapters

*Answer: A. The post says "I have experienced occasional to frequent loss in signal when listening to stations that utilize TuneIn."*

8. **When discussing Matter devices going offline, which brand of mesh router system was singled out for Thread settings confusion?**

   A. Eero 6E

   B. TP-Link Deco

   C. UniFi Dream Router

   D. Netgear Orbi

   *Answer: A. A user wrote "I have an Eero 6e mesh router system... The Threads feature is toggled on," yet their Matter gear still dropped.*

9. **How did a Nest Mini owner describe the music delay when the speaker was added to a stereo link in Google Home?**

   A. The delay is HUGE.

   B. It syncs perfectly.

   C. Only milliseconds of lag.

   D. Delay happens once a month.

   *Answer: A. The post states, "If I play any music... the delay is HUGE," emphasizing a negative feeling about latency.*

10. **Which future-oriented speculation did shoppers raise after noticing no Nest Audio stock in multiple country stores?**

    A. A new generation might be announced at the Pixel event

    B. Google is switching to Apple HomeKit

    C. All smart speakers will become subscription-based

    D. Wi-Fi will be removed from Nest

    *Answer: A. They asked, "Are people expecting a new generation to be announced at the Pixel event in a couple weeks?"—a tendency toward anticipating new hardware.*

11. **Which cloud storage dilemma did dual-ecosystem users discuss while already owning many Nest Hubs and iCloud devices?**

    A. Paying for both 200 GB iCloud and 200 GB Google One plans

    B. Choosing between Dropbox and Box free tiers

    C. Losing access to Microsoft OneDrive photos

    D. Migrating from Amazon S3 Glacier Vaults

    *Answer: A. The repeated post describes both iCloud and Google One hitting the 200 GB limit and not wanting to upgrade both.*

12. **What network feature on apartment Wi-Fi prevented an elderly resident's Nest Mini from completing setup?**

    A. AP Isolation turned on

    B. Hidden SSID broadcast

    C. WPA3-Enterprise only

    D. Dual NAT tunneling

    *Answer: A. The care home enables "AP Isolation," so the speaker throws the message "Please check your Wi-Fi network settings."*

13. **Which sound-related improvement motivated users to prefer Nest Audio over their old Google Home Minis?**

    A. 'Bass is the most noticeable improvement' at high volume

    B. Built-in CD player support

    C. Dolby Atmos rear channels

    D. Quad-mic noise cancelling

    *Answer: A. One review says, "Bass is the most noticeable improvement, high volume performance is better," highlighting the aspect of audio quality.*

14. **How much did Canadian bargain hunters report paying at Lowe's or Home Depot for clearance Nest Audio units?**

    A. $39.97

    B. $129.99

    C. $199.00

    D. $15.00

    *Answer: A. Posts note "Nest Audio for sale for $39.97... is it worth getting," reflecting pricing sentiment.*

15. **Which workaround did some owners adopt because the Nest Hub could no longer resume music on the intended room speaker?**

    A. Using the broadcast command instead of TTS

    B. Switching to Zigbee bulbs

    C. Turning on microphone sensitivity

    D. Downgrading firmware via USB

    *Answer: A. One poster said they had to "resort to using broadcast commands which are clunky" when TTS stopped working.*

16. **What phrase did a Nest thermostat user shout after eco mode kept activating despite settings being disabled?**

    A. "Jeezus Google."

    B. "Bravo Assistant!"

    C. "Mission accomplished!"

    D. "Danke Alexa."

    *Answer: A. The frustrated quote is "Jeezus Google," showing irritation with unwanted eco behaviour.*

17. **When debating cloud versus local AI, which low-power device did a homeowner consider dedicating as an "always-on screen" for NotebookLM?**

    A. An old MacBook Pro

    B. A Raspberry Pi Zero

    C. A Lenovo Tab M8

    D. A Pixel 2 phone

    *Answer: A. They planned "to dedicate an old MacBook Pro for the AI assistant" but were open to tablets.*

18. **Which free radio service did a listener compare to TuneIn, noting that Audacy retained signal 'much better' on Nest speakers?**

    A. Audacy

    B. Pandora

    C. SiriusXM

    D. iHeartWeather

    *Answer: A. The poster said "The other services like Audacy work much better in terms of signal retention," offering a comparison.*

19. **What is one reason a user said HomePods were a 'let down' compared with Google Nest speakers?**

    A. 'You have to AirPlay from your phone' for many tasks

    B. They lack any bass at all

    C. They cannot set alarms

    D. HomePods don't support Wi-Fi

    *Answer: A. The criticism: "on HomePod minis you have to air play from your phone... google are just way ahead," indicating a comparison and feeling.*

20. **Faced with a 2025 Nest price hike e-mail, how many days of event video history does the 'Nest Aware Plus' plan promise to keep?**

A. 60 days

B. 30 days

C. 7 days

D. 120 days

*Answer: A. The notice states the plan keeps "60 days of event video history... up to 10 days of 24/7 video history," summarising the aspect of subscription value.*

### A.4.3 GOOGLE NEST SMART SPEAKER QUESTIONNAIRE FROM QWEN3-30B-A3B

1. **In your smart home setup, how do you integrate Google Nest Mini speakers into an older home intercom system?**

   A. By replacing the entire intercom system with new wireless speakers

   B. By using a 24VDC power supply and buck converters to repurpose old wiring

   C. By connecting the speakers directly to a home Wi-Fi network without rewiring

   D. By mounting the speakers to the wall using adhesive pads only

   *Answer: B. By using a 24VDC power supply and buck converters to repurpose old wiring*

2. **What is the primary reason for using Google Nest Audio speakers in a stereo setup for music playback?**

   A. They offer better sound quality and deeper bass than other smart speakers

   B. They are the only Google devices compatible with Apple Music

   C. They support Bluetooth audio output without delay

   D. They are designed specifically for use with Sonos systems

   *Answer: A. They offer better sound quality and deeper bass than other smart speakers*

3. **When using Google Nest Mini as a temporary speaker during a move, what feature is most important for portability?**

   A. Built-in battery and rechargeable power source

   B. Ability to connect via USB-C to a TV

   C. Support for HDMI input from a computer

   D. Integration with Apple HomeKit for remote access

   *Answer: A. Built-in battery and rechargeable power source*

4. **Why might a user choose Google Nest Mini over Apple HomePod Mini for a smart home setup?**

   A. Because Apple HomePod Mini requires a HomePod Hub for full functionality

   B. Because Google Nest Mini supports more third-party integrations and voice commands

   C. Because HomePod Mini lacks music playback capabilities

   D. Because Google Mini is compatible with all smart home ecosystems

   *Answer: B. Because Google Nest Mini supports more third-party integrations and voice commands*

5. **What has caused frustration for users who rely on Google Nest speakers for music playback and voice commands?**

   A. Frequent unresponsiveness and inconsistent command recognition

   B. Inability to support Spotify Premium accounts

   C. Limited compatibility with Bluetooth devices

   D. High cost of replacing outdated models

   *Answer: A. Frequent unresponsiveness and inconsistent command recognition*

6. **How do users perceive the reliability of Google Nest devices after the company's shift to Gemini AI?**

   A. They believe the new AI will improve performance and voice recognition

   B. They are concerned about service degradation and potential obsolescence

    C. They expect all devices to be replaced within a year

    D. They trust that Google will maintain full support indefinitely

*Answer: B. They are concerned about service degradation and potential obsolescence*

7. **What do users suggest as the best way to maintain a smart home ecosystem without relying on Google's services?**

    A. Using Home Assistant with a local server for full control

    B. Switching to Apple HomeKit for better integration

    C. Purchasing only Amazon Echo devices for compatibility

    D. Replacing all devices with non-smart alternatives

*Answer: A. Using Home Assistant with a local server for full control*

8. **What is a common workaround when a Google Nest speaker fails to connect to Wi-Fi during setup?**

    A. Changing the account region to the United States in the app settings

    B. Using a wired Ethernet connection instead of Wi-Fi

    C. Factory resetting the device and re-registering it

    D. Replacing the speaker with a newer model

*Answer: A. Changing the account region to the United States in the app settings*

9. **Why do some users prefer Google Nest Audio over other smart speakers for music listening?**

    A. It has superior stereo sound quality and bass response

    B. It is the most affordable smart speaker with no subscription fees

    C. It supports more music streaming services than any other model

    D. It integrates with Google Photos for visual album displays

*Answer: A. It has superior stereo sound quality and bass response*

10. **What is the main concern for users who are invested in both Apple and Google ecosystems?**

    A. Managing duplicate cloud storage between iCloud and Google One

    B. Choosing between Apple HomePod and Google Nest for smart home control

    C. Upgrading to new smartphones to maintain compatibility

    D. Finding compatible smart bulbs for both platforms

*Answer: A. Managing duplicate cloud storage between iCloud and Google One*

11. **How do users feel about the future of Google Nest hardware following recent service issues?**

    A. They believe Google will soon release a new generation of devices

    B. They are skeptical about long-term support and product continuity

    C. They expect Google to fully open-source the software for user maintenance

    D. They are confident that all devices will continue to function indefinitely

*Answer: B. They are skeptical about long-term support and product continuity*

12. **What challenge arises when trying to control Google Nest devices from a non-Google ecosystem device?**

    A. Voice commands are not recognized due to language mismatch

    B. The speaker cannot detect voice input from Apple devices

    C. Integration with Home Assistant becomes unreliable or impossible

    D. The device refuses to connect to Wi-Fi networks outside the Google network

*Answer: C. Integration with Home Assistant becomes unreliable or impossible*

13. **What is a key limitation when using Google Nest Mini speakers for doorbell alerts?**

    A. The alert only plays on one speaker, not all connected devices

    B. The doorbell chime only works when the speaker is unplugged

    C. The speaker cannot broadcast alerts during music playback

    D. Alerts are delayed by 10–15 seconds due to app routing

*Answer: A. The alert only plays on one speaker, not all connected devices*

14. **Why might a user consider replacing a Google Nest Hub Max despite its popularity?**

    A. It is no longer available for purchase in many regions

    B. It has a built-in camera that cannot be disabled

    C. It requires a paid subscription for basic functionality

    D. It only works with Apple devices and not Android

*Answer: A. It is no longer available for purchase in many regions*

15. **How do users describe the current performance of Google Nest Hub devices in 2025?**

    A. They are slow, glitchy, and frequently misinterpret commands

    B. They have improved dramatically with the new Gemini update

    C. They function as well as they did five years ago

    D. They are only useful for playing music and showing photos

*Answer: A. They are slow, glitchy, and frequently misinterpret commands*

16. **What is a major reason users are hesitant to expand their Google Nest ecosystem?**

    A. Fear of being locked into a system that may no longer be supported

    B. High cost of additional smart home accessories

    C. Incompatibility with older Wi-Fi routers

    D. Lack of support for non-English languages

*Answer: A. Fear of being locked into a system that may no longer be supported*

17. **What is the most common solution users suggest for improving audio performance across multiple Nest speakers?**

    A. Upgrading to a higher-end model like the Google Home Max

    B. Using a dedicated audio amplifier for better sound quality

    C. Replacing all speakers with Sonos devices for consistent playback

    D. Using a local server to manage audio streams and reduce latency

*Answer: D. Using a local server to manage audio streams and reduce latency*

18. **What is a major drawback of using Google Nest Audio as a TV speaker?**

    A. It has a significant audio delay when connected via Bluetooth

    B. It lacks a 3.5mm input port for direct audio connection

    C. It cannot play video content from streaming apps

    D. It only works with Google TV devices

*Answer: A. It has a significant audio delay when connected via Bluetooth*

19. **What do users suggest as an alternative to Google Nest speakers for a private, local smart home system?**

    A. Using a Raspberry Pi with a local LLM for voice control

    B. Replacing all devices with Amazon Echo Dot models

    C. Switching to Apple HomePods for better privacy

    D. Using only non-smart speakers with manual controls

*Answer: A. Using a Raspberry Pi with a local LLM for voice control*

20. **How do users perceive the value of older Google Nest devices like the Nest Mini and Nest Audio?**

    A. They are still functional and affordable, especially when bought secondhand

    B. They are outdated and no longer supported by Google

    C. They are only useful for basic tasks like playing alarms

    D. They are incompatible with modern Wi-Fi networks

*Answer: A. They are still functional and affordable, especially when bought secondhand*

A.5   Lighted Tree of the questionnaires on Google Nest Smart Speaker

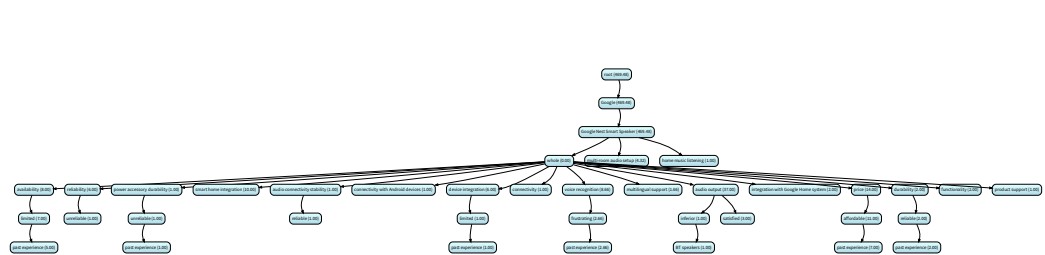

Figure 4: Lighted Tree from GPT-5 on Google Nest Smart Speaker.

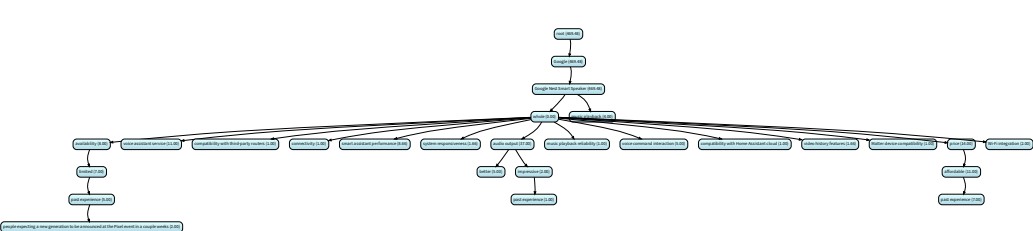

Figure 5: Lighted Tree from GPT-o3 on Google Nest Smart Speaker.

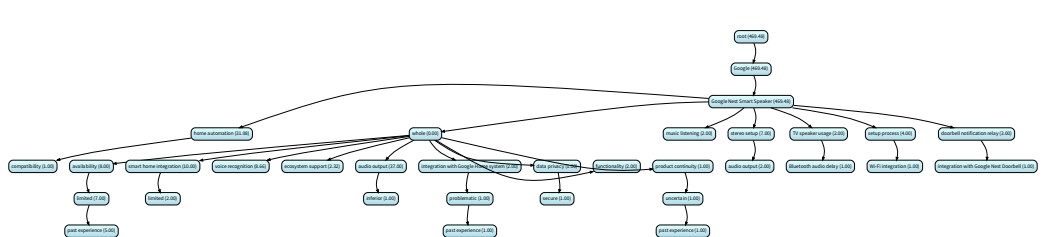

Figure 6: Lighted Tree from Qwen3-30B-A3B on Google Nest Smart Speaker.

