# OpenReview forum: "CONSINTBENCH: EVALUATING LANGUAGE MODELS ON REAL-WORLD CONSUMER INTENT UNDERSTAND- ING"
_ICLR.cc/2026/Conference — ICLR 2026 Conference Withdrawn Submission_

### Official Review · Reviewer_FbLv · 2025-10-25

**Soundness:** 1
**Presentation:** 3
**Contribution:** 2
**Rating:** 4
**Confidence:** 4

**Summary:**

This paper proposes CONSINTBENCH, a benchmark for evaluating large language models in understanding real-world consumer intent. The dataset contains over 200,000 product discussions across nine major domains, 54 subdomains, and over 1,400 products. The paper defines four evaluation dimensions: depth (divided into five layers: L1-L3 for basic understanding, L4-L5 for advanced reasoning), breadth, informativeness, and correctness. Methodologically, the paper constructs CONSINT-TREE (a hierarchical tree-structured knowledge graph) to assess depth and breadth, CONSINT-RAG (retrieval-augmented generation) to assess correctness, and informativeness through lexical richness and semantic redundancy. The evaluation process adopts the LLM paradigm of self-generated questionnaires and answers. Experiments evaluate 20 models, including the GPT series, Claude, the Qwen series, and LLaMA. Results show that reasoning models outperform general-purpose models in most dimensions, but all models still lag significantly behind in understanding deep intent. Ablation experiments show that using CONSINT-TREE instead of the original discussion can improve the depth and breadth scores but reduce the accuracy.

**Strengths:**

1. This paper identifies a critical area of real-world multi-user, multi-perspective intent understanding that has been overlooked by existing research, and accurately analyzes the limitations of existing benchmarks that rely on synthetic data.

2. This paper constructs a large-scale dataset containing over 200,000 real consumer discussions, covering over 1,400 products and 54 sub-domains across nine major domains. The automated data collection pipeline supports dynamic updates, effectively preventing data contamination.

3. The proposed four-dimensional evaluation system (depth, breadth, informativeness, and correctness) is reasonable, especially the breakdown of depth into five levels, providing a clear design approach from basic understanding to advanced reasoning.

4. This paper evaluates 20 mainstream models, including the latest GPT-5 and GPT-o3, across various scales (1.5B to 72B) and types (both reasoning and general-purpose models), with a comprehensive experimental setup.

5. The experimental results reveal the shortcomings of current state-of-the-art models in deep intent understanding, with all models scoring close to zero at the L4-L5 levels. This suggests a path for improvement in future research.

**Weaknesses:**

1. I understand that the authors' adoption of a self-generated evaluation paradigm aims to comprehensively assess the model's ability to grasp intent, and this design approach is reasonable. However, I believe that this approach of having the LLM generate questions, answer them, and then evaluate its own answers carries a potential risk: the model may favor questions it excels at answering, resulting in high scores, but without actually gaining a deep understanding of consumer intent. The results in Table 4 seem to confirm this concern: Qwen3-30B-A3B scores significantly higher than GPT-5 on L1, but the authors themselves point out that its questions are less informative and more verbose, suggesting that the model may be overwriting the tree structure by generating a large amount of shallow content. I believe this issue may affect the core validity of the benchmark, as high scores may reflect the model's verbosity rather than its true depth of understanding.

2. I note that the paper relies entirely on automatic evaluation, without any human validation. I understand that large-scale human evaluation is costly, but as a newly proposed benchmark, I consider the lack of human validation a significant limitation. This makes it difficult to determine whether CONSINT-TREE's five-layer structure truly reflects human understanding of the hierarchy of intent, nor whether the automatic scoring results are consistent with the judgments of domain experts. For example, when GPT-o3 scores 0.07 on L5, I can't tell whether this is because the model truly lacks understanding of deep intent or because the evaluation method is not sensitive enough to capture such capabilities. Without human evaluation, the validity and credibility of the benchmark cannot be fully guaranteed.

3. I believe the design of the evaluation pipeline may introduce the risk of systemic bias. CONSINT-TREE relies on the GPT-4o extraction branch, and CONSINT-RAG's judgment also uses the LLM. The entire evaluation criteria is heavily based on the semantic understanding of a specific model. While this is feasible in engineering implementation, I worry that it may bias the evaluation results towards models with similar expression styles to GPT-4o. Different models may express the same intent in different ways, but if the evaluation criteria are primarily based on GPT-4o's understanding framework, models that express differently but correctly understand intent may be underestimated. This potential bias may affect the fairness of the benchmark as a general evaluation standard.

4. I found the paper lacking in completeness of technical details. While Appendix A.2 provides some explanation, key information such as the similarity threshold for node merging in the CONSINT-TREE construction, the specific parameters of the matching mechanism, and the selection of top-k values in CONSINT-RAG remain unclear. The prompt design is completely absent. The choice of these hyperparameters can significantly affect the evaluation results, but the paper does not discuss the rationale for these choices or provide ablation analysis. The lack of complete algorithm pseudocode also makes it difficult for other researchers to accurately replicate the work. I believe that for a benchmark that aspires to become a community standard, these missing technical details are necessary.

5. I believe the paper could improve both the depth and breadth of the experimental analysis. The ablation experiments in Table 3 were conducted on only three models, resulting in a small sample size, making it difficult to conclude that the with/without tree approach is universally effective. More importantly, the paper primarily reports experimental results and lacks in-depth discussion of the underlying reasons. For example, why does the with-tree approach improve depth and breadth but reduce accuracy? What cognitive deficits lead to the poor performance of all models at the L4-L5 levels? Does this task involve higher-level cognitive abilities such as Theory of Mind or causal reasoning? As a research paper, I think these in-depth analyses and theoretical insights are important, but the current paper does not discuss this aspect sufficiently.

**Questions:**

1. How do the authors address the concern that the self-generated evaluation paradigm may measure model consistency rather than true intent understanding? Have they considered using a fixed set of expert-designed questions for all models to answer, allowing for a more direct comparison of different models' understanding abilities? Alternatively, are there other validation methods to ensure that the current evaluation truly captures intent understanding rather than model verbosity?
2. Can human evaluation experiments be supplemented to verify the effectiveness of the automatic evaluation? Even on a relatively small sample size, calculating inter-annotator agreement and correlation between human and machine scores would greatly enhance the credibility of the benchmark. If human-machine correlation is low, how will the authors adjust their evaluation methods?

3. How do they ensure that the evaluation pipeline is fair to models with different expressive styles? Have they attempted to construct CONSINT-TREE using different LLMs (e.g., Claude or open-source models) and compare the results? If different construction methods lead to significantly different evaluation results, what does this indicate? Is it possible to adopt a more model-neutral construction method?

4. Can you provide complete technical details, including algorithm pseudocode, specific values of key hyperparameters and their selection rationale, and prompt design for key steps? Can you conduct ablation experiments on hyperparameters to demonstrate the impact of different parameter choices on the evaluation results? This information is crucial for the community to replicate and improve the work.

5. Can the findings from the ablation experiments be validated on more models to confirm their generalizability? More importantly, what insights do the authors have into the cognitive mechanisms underlying these experimental results? Why do models generally perform poorly at L4-L5? What capabilities does this reflect? Based on these findings, can you offer hypotheses or suggestions for model improvement?

---

### Official Review · Reviewer_Tbc9 · 2025-10-30

**Soundness:** 2
**Presentation:** 2
**Contribution:** 2
**Rating:** 2
**Confidence:** 4

**Summary:**

The paper describes a live benchmark of product discussions and a framework for evaluating LLM understanding of consumer intent. The benchmark covers a number of domains and 1400+ products. Data is updated daily to avoid training contamination through an automatic retrieval and cleaning process. A multidimensional evaluation combining rule-based and LLM-judges covers depth and breadth of understanding, correctness and informativeness of responses. Analysis of 20 contemporary open-source and proprietary LLMs is provided.

**Strengths:**

The paper provides a new benchmark for analyzing product discussions from a variety of angles. The development and maintenance of a live benchmark is admirable.

**Weaknesses:**

I struggled to understand from the paper how the depth of models are evaluated. From what I can tell, models under evaluation are asked to generate a questionnaire with answers from the raw discussion, and then these questionnaires are compared to a structured tree generated by GPT-4o which is prompted in a specific way to extract “deeper” levels of conversation analysis. If this is true, this biases the evaluation in favor of GPT models since a GPT analysis is treated as the gold truth. It also seems to not accurately evaluate model capability, since creating a questionnaire and providing deep analysis of consumer intent are different tasks. If I have misunderstood the depth evaluation, it is because the paper did not clearly describe the exact process of this evaluation.


Overall, the over-reliance on automatic evaluation in this benchmark detracts from its usefulness. In addition to the above mentioned bias, other evaluations such as Infomativeness rely on weak lexical measures and vector similarity. The paper provides no evidence that these correspond to human notions of informativeness.

**Questions:**

What is the prompt for creating questionnaires?


Using the tree-generation prompt, what is the agreement between strong models from different families on different levels of analysis?

---

### Official Review · Reviewer_q6SU · 2025-11-01

**Soundness:** 2
**Presentation:** 2
**Contribution:** 2
**Rating:** 4
**Confidence:** 3

**Summary:**

The paper introduces CONSINT-Bench, a benchmark for evaluating how well LLMs understand real-world consumer intent. It aggregates roughly 200K opinions from open websites covering more than 1,400 products and domains. The evaluation framework includes a five-level hierarchical CONSINT-Tree to score the depth and breadth of inferred intents, and CONSINT-RAG that evaluates answer accuracy against source discussions and quantifies informativeness.

**Strengths:**

The paper targets an important goal, evaluating how well LLMs understand real world consumer intent. The proposed five-level CONSINT-Tree is an interesting, structured design that tries to capture progressively richer signals (from basic usage scenarios and product aspects to comparative trade-offs and tendencies), offering more interpretable depth and breadth measurements than flat metrics.

**Weaknesses:**

1. The benchmark design is confusing: the paper aims to measure customer intent understanding with LLMs and introduces a five-level CONSINT-Tree for “depth,” yet it provides no empirical evidence that this taxonomy is a valid proxy for customer intent. Because “intent” is inherently abstract, the paper should offer an operational definition and demonstrate validity. A common approach to evaluate the customer intent is to predict the shoppers actions, such as purchase or not purchase behavior (e.g. Shop-R1). Without such evidence, the link between the stated motivation (understanding customer intent) and the chosen design (tree coverage and RAG-based correctness) is very weak.
2. The writing is difficult to follow, and the benchmark design is not explained clearly. In addition, releasing the benchmark (and code) on huggingface will be helpful to evaluate the data quality.

Related work:
Shop-R1: https://arxiv.org/pdf/2507.17842

**Questions:**

The authors should provide strong evidence that their design is a good proxy of customer intent, for example it can be used to predict the customer behavior.

---

### Official Review · Reviewer_tSqV · 2025-11-01

**Soundness:** 2
**Presentation:** 2
**Contribution:** 2
**Rating:** 2
**Confidence:** 3

**Summary:**

The paper proposes CONSINT-BENCH, a benchmark designed to evaluate LLMs’ ability to understand real-world consumer intent. It introduces four evaluation dimensions: depth, breadth, informativeness, and correctness, and two corresponding evaluation frameworks: CONSINT-TREE, which measures the depth and breadth of intent understanding, and CONSINT-RAG, which evaluates correctness and informativeness.

**Strengths:**

The motivation is good; understanding consumer intent is an important and I haven't seen a similar dataset yet. The benchmark also covers diverse consumer-related categories, ranging from vehicles to personal care items. In addition, the four evaluation dimensions offer a broad perspective on assessing LLMs’ intent-understanding abilities.

**Weaknesses:**

This paper is difficult to fully understand in terms of what it is actually proposing. The overall task definition and experimental setup are unclear.
The paper does not provide any examples of data points or clearly describe where the data come from. It is unclear what kinds of consumer discussions were collected, how user intents were extracted, or what specific types of intent the benchmark aims to measure.

The introduction of CONSINT-TREE and CONSINT-RAG adds further confusion. From my understanding, CONSINT-TREE is meant to represent user intent across multiple levels of depth, while CONSINT-RAG is a retrieval-augmented framework for evaluating correctness by comparing model outputs against retrieved human opinions. However, it is ambiguous what the LLMs are supposed to predict, what the exact inputs are, and what constitutes relevant or irrelevant information in this benchmark.
Overall, the paper fails to adequately describe the dataset construction process and data quality before introducing the technical components such as CONSINT-TREE and CONSINT-RAG.

Lastly, the tree-based evaluation is also confusing. The tree diagrams seem to contain only single-word nodes at different depths, making it unclear how this structure captures or quantifies "intent understanding". The rationale for using such a hierarchical tree as an evaluation metric is not well justified.

**Questions:**

Figure 2 is too small to read.

---

### Note · Authors · 2026-01-08

I have read and agree with the venue's withdrawal policy on behalf of myself and my co-authors.